# QUATgo: Protein quaternary structural attributes predicted by two-stage machine learning approaches with heterogeneous feature encoding

Chi-Hua Tung[1]☯, Ching-Hsuan Chien[2,3]☯, Chi-Wei Chen[3,4], Lan-Ying Huang[3], Yu-Nan Liu[3], Yen-Wei Chu[3,5,6,7,8,9]*

1 Department of Bioinformatics, Chung-Hua University, Hsinchu, Taiwan (R.O.C.), 2 Ph.D. Program in Medical Biotechnology, National Chung Hsing University, Taichung City, Taiwan (R.O.C.), 3 Institute of Genomics and Bioinformatics, National Chung Hsing University, Taichung City, Taiwan (R.O.C.), 4 Department of Computer Science and Engineering, National Chung Hsing University, Taichung City, Taiwan (R.O.C.), 5 Institute of Molecular Biology, National Chung Hsing University, Taichung City, Taiwan (R.O.C.), 6 Agricultural Biotechnology Center, National Chung Hsing University, Taichung City, Taiwan (R.O.C.), 7 Biotechnology Center, National Chung Hsing University, Taichung City, Taiwan (R.O.C.), 8 Ph.D. Program in Translational Medicine, National Chung Hsing University, Taichung City, Taiwan (R.O.C.), 9 Rong Hsing Research Center For Translational Medicine, Taichung City, Taiwan (R.O.C.)

☯ These authors contributed equally to this work.
* ywchu@nchu.edu.tw

**Data Availability Statement:** QUATgo is freely accessible to the public as a web server via the site at http://predictor.nchu.edu.tw/QUATgo.

## Abstract

Many proteins exist in natures as oligomers with various quaternary structural attributes rather than as single chains. Predicting these attributes is an essential task in computational biology for the advancement of proteomics. However, the existing methods do not consider the integration of heterogeneous coding and the accuracy of subunit categories with limited data. To this end, we proposed a tool that can predict more than 12 subunit protein oligomers, QUATgo. Meanwhile, three kinds of sequence coding were used, including dipeptide composition, which was used for the first time to predict protein quaternary structural attributes, and protein half-life characteristics, and we modified the coding method of the functional domain composition proposed by predecessors to solve the problem of large feature vectors. QUATgo solves the problem of insufficient data for a single subunit using a two-stage architecture and uses 10-fold cross-validation to test the predictive accuracy of the classifier. QUATgo has 49.0% cross-validation accuracy and 31.1% independent test accuracy. In the case study, the accuracy of QUATgo can reach 61.5% for predicting the quaternary structure of influenza virus hemagglutinin proteins. Finally, QUATgo is freely accessible to the public as a web server via the site http://predictor.nchu.edu.tw/QUATgo.

## Introduction

A considerable number of proteins actually consist of numerous polypeptide chains assembled together. That is, rather than existing as individual chains, a large number of proteins take the

**Funding:** This research was supported by the Ministry of Science and Technology, Taiwan, R.O. C. [grant numbers 106-2221-E-005-077-MY2, 107-2634-F-005-002, and 107-2321-B-005 -013], and the National Chung Hsing University and Chung-Shan Medical University [grant number NCHU-CSMU-10705]. The funders had no role in study design, data collection and analysis, decision to publish, or preparation of the manuscript.

**Competing interests:** The authors have declared that no competing interests exist.

form of oligomers in vivo, with these oligomers having a variety of quaternary structural characteristics. These oligomers serve as the structural bases for a range of biological functions, including ion-channel gating, allosteric mechanisms, and various cooperative effects [1]. The oligomers have a fourth-level structure termed the quaternary structure, even as the majority of proteins only consist of a primary structure (i.e., an amino acid sequence), a secondary structure (i.e., alpha-helices and beta-sheets), or a tertiary structure such as a structure with three-dimensional folding). In any case, the term "quaternary structure" refers to both the number of protein subunits and the arrangement of those subunits with one another. Ion channel proteins, hemoglobin, and DNA polymerase are just a few examples of proteins that have quaternary structures [2]. As indicated above, the protein quaternary structure is itself composed of more than one tertiary structure, with the quaternary structure being part of the protein's overall three-dimensional structure, and X-ray crystallography or NMR studies can be used to determine a given quaternary structure. Relatedly, in the field of structural bioinformatics, the prediction of the quaternary structural characteristics of proteins plays a critical role.

As far as we know, the earliest experiments to study the type of protein quaternary structure were in 2001 [3]. In a study in 2003, Garian et al. and Zhang et al. [4] used a support vector machine (SVM) as a classifier to identify whether an unknown sequence was a homo-dimer by using the random amino acid composition and integrating the AAindex as a feature to predict the quaternary structure of proteins. These studies confirmed the importance of protein secondary structure for quaternary structures, and Zhang et al. also used virtual PseAAC to improve the accuracy of quaternary structure prediction. Then, in 2006, Shi et al. [5] predicted homo-oligomers based on the amino acid composition distribution, also known as AACD, and proposed a two-dimensional principal component analysis. A good method called 2DPCA can effectively solve high-dimensional special vectors. Levy proposed the PiQSi database, which took 15,000 annotated protein quaternary structure sequences from the PDB as a benchmark dataset to test the accuracy of different methods for predicting the quaternary structure of proteins [6]. In 2009, Xiao et al. proposed a 2-layer predictor to predict protein quaternary structure and constructed a prediction website called Quat-2L [7]. In recent years, Shen et al. have also proposed coding methods based on functional domain composition [1]. Protein functional domains are related to molecular evolution. Such domains are used as building blocks and reorganized in different arrangements to regulate protein function. Proteins usually consist of multiple functional domains, so we define the conserved domain as a repeat unit in molecular evolution, and its range can be determined by sequence and structure analysis. If the protein sequence has similar functional domains, it may represent an evolutionarily relationship [8, 9]. Comparing similarities can be performed with RPS-BLAST through the Conserved Domain Database (CDD) [10]. This method has also been confirmed to improve the accuracy of predicting quaternary structures, but the disadvantage is that some proteins may not match the functional domains in the database. This may be caused by the fact that the database is not yet complete; defining each functional domain as a different dimension may make the feature vector too large. The inspection algorithm causes too much noise in the decision-making process. In 2012, Sun et al. used the discrete wavelet transform based on Chou's PseAAC to identify the quaternary structure properties of proteins [11]. These abovementioned methods are based on a set of features and are also two-tier system architectures [12].

The purpose of constructing a quaternary structure prediction system is to quickly identify the type of oligomeric protein from unknown protein sequences. For proteins of different functions, sources and species, in addition to verifying structural information, it is more difficult to find the protein aggregation rules using the alignment method if the homology between

the sequences is low. Therefore, it is necessary to develop a prediction system that can provide users with quick and reliable answers [13]. The experimenter often obtains the expected protein sequence through amino acid sequencing. The resulting sequence is the amino acid sequence of the intact protein rather than the sequence registered in UniProt, so based on the experimenter's point of view, the input sequence provided by the system is a complete protein sequence. We also consider the concept of protein interactions, use the pseudo amino acid composition, linking the protein-binding surface and the solvent accessibility to calculate the sequence of amino acids exposed on the surface of the protein and calculated the amino acid hydrophilicity and hydrophobicity value of the protein, which has an important influence on folding and interaction. We added the encoding method that is used for the first time in protein interaction, including the dipeptide composition that has been shown to improve the accuracy of the protein secondary structure prediction. Also, we have modified the coding method of functional domain composition, which has reduced the problem regarding the feature vector being too large, and this is the first time the half-life prediction (HLP) has been used to predict the issues of protein quaternary structure prediction. HLP is a characteristic coding that was used to predict the half-life of proteins in the small intestine environment. Peptides have been suggested to be good drug candidates, and their usage is still hindered mainly because of their high susceptibility towards protease degradation. Therefore, peptides are cited as a coding feature of the four-level structure regarding this topic.

The existing systems are based on a set of features, and they are also two-layer system architectures. QUATgo is a two-stage prediction model with 10-fold cross validation. The first stage prediction model separates each subunit of a homologous hetero-oligomer into 16 subsets and uses the random tree forest algorithm to perform model training, which uses a single model to obtain 16 results. Because the number of hetero-decamer is too small, category X is defined together with the hetero- more than dodecamer, and we obtain 49.0% cross-validation accuracy and 31.1% independent test accuracy. If the predicted result is category X, the system will send the data to the second stage prediction model to further distinguish whether the sequence is hetero-decamer or hetero- more than dodecamer. The second stage of the prediction model uses the support vector machine as the algorithm and can achieve 100% accuracy in cross-validation and 95.9% accuracy in independent testing. The entire system will eventually have 49.0% cross-validation accuracy and 31.1% independent test accuracy. To verify whether the system has sufficient predictive ability in biological research on other topics, we have made a prediction of hemagglutinin proteins of influezna A and B. QUATgo can reach 61.5% of accuracy.

## Materials and method

### Dataset preparation

The data for the oligomeric protein sequences were obtained from 3D Complex (http://www.3dcomplex.org/) [14] and PiQSi (http://www.PiQSi.org/) [6]. Each downloaded FASTA sequence was sent to the Protein Data Bank (PDB) (https://www.rcsb.org/) for protein quaternary structure confirmation, and the dataset was subdivided into 9 subcategories according to the number of subunits (monomer, dimer, trimer, tetramer, hexamer, octamer, decamer, dodecamer, more than dodecamer). The protein sequence was then processed using CD-HIT [15] for acquaintance processing. If the sequence number of the subset was more than 2,000, the sequence with similarity greater than 50% was eliminated. If the sequence number of the subset was between 200 and 2,000, the sequence with similarity greater than 60% was eliminated for reducing module learning with high similarity data and the system operating time. If the number of the same oligosaccharide protein sequence data is less than 200, no similarity

Table 1.  Data numbers of each subunit before and after CD-Hit processing.

| | Before CD-hit | | After CD-hit | |
|---|---|---|---|---|
| | **Homomer** | **Heteromer** | **Homomer** | **Heteromer** |
| Monomer | 10461 | 10461 | 1815 | 1815 |
| 2 mer | 5995 | 993 | 2115 | 278 |
| 3 mer | 747 | 245 | 312 | 96 |
| 4 mer | 1907 | 486 | 655 | 156 |
| 5 mer | 64 | 7 | 63 | 7 |
| 6 mer | 474 | 91 | 201 | 91 |
| 7 mer | 20 | 9 | 20 | 9 |
| 8 mer | 170 | 51 | 170 | 51 |
| 9 mer | 0 | 32 | 0 | 32 |
| 10 mer | 53 | 20 | 53 | 20 |
| 11 mer | 4 | 1 | 4 | 1 |
| 12 mer | 98 | 78 | 98 | 78 |
| mt12 mer* | 74 | 79 | 74 | 79 |
| Total | 20067 | 12553 | 5580 | 2713 |

* = more than 12 mer

removal will be carried out to avoid the loss of sufficient statistical significance due to the low number of machine learning data. The results of CD-HIT are presented in Table 1. The system could not predict oligomers of 5-mer, 7-mer, 9-mer, or 11-mer because they are too small for training data.

## Encoding method

**Amino acid base.** The amino acid base coding method was based on the binary compilation method, which divides 20 common amino acids into 20 different vector dimensions. The purpose was to distinguish the differences in the actual physicochemical properties of the amino acids. It was not possible to represent the differences in physicochemical properties. Therefore, 20 common amino acids were defined in different dimensions, and the frequency and occurrence of each amino acid in the sequence were calculated.

Dipeptide feature composition is an important parameter of protein structure, and previous experiments have confirmed that it can improve the accuracy of protein secondary structure prediction [16], construct a 400-dimensional feature vector with bi-amino acid composition, and calculate dipeptide groups with its frequency and number of occurrences of acid in the sequence [17, 18].

**Amino acid index.** AAindex is a database of numerical indices representing various physicochemical and biochemical properties of amino acids and pairs of amino acids. Another important feature of amino acids that can be represented numerically is the similarity between them. Thus, a similarity matrix, also called a mutation matrix, is a set of 20 x 20 numerical values used for protein sequence alignments and similarity searches. The AAindex consists of three parts: AAindex1 represents the 20-value amino acid index, AAindex2 represents the amino acid mutation matrix, and AAindex3 represents the statistical protein contact potential. All data come from published literature [19–22]. Twenty AAindex indices were used in this study, which included side chain orientational preference, isoelectric point, ratio of average and computed composition, polarizability parameter, steric parameter, solvation free energy, normalized frequency of turn, normalized frequency of beta-sheet, normalized frequency of

beta-turn, normalized frequency of alpha-helix, hydrophobicity index, flexibility parameter, transfer free energy to surface, alpha-NH chemical shifts, alpha-CH chemical shifts, polarity, secondary structure, molecular size or volume, codon diversity, and electrostatic charge. The encoding method was based on the probability of amino acid occurrence in the sequence multiplied by the index.

**Functional Domain (FunD) composition.** The Conserved Domains Database [23] is a part of NCBI's Entrez query and retrieval system for providing the conserved functional domains of annotated protein sequences. Most proteins contain more than one functional domain, which represents highly similar structures and functions. We can derive whether the proteins may have evolved from the same ancestor from the results of these proteins, and such highly conserved characteristics become important clues to study protein functions or related mechanisms. In contrast to previous research that defined the protein feature as a 50369-dimensional vector, we define each database as a number of feature dimensions and encode the resulting E-value and Bitscore as the average value. We also calculate the number of the three hit types (specific, non-specific, superfamily) of the CDD searching results in this study. The purpose of reducing the space vector was to decrease the system operation time, and a large number of feature dimensions may cause noise in the prediction results.

**Parallel correlation pseudo amino acid composition.** The parallel correlation pseudo amino acid composition (PC-PseAAC) under Pse-in-One (http://bioinformatics.hitsz.edu.cn/Pse-in-One/home/) was used [24, 25]. Pse-in-One is a web-based tool that uses virtual components and different virtual amino acid calculation formulas to calculate user-entered DNA, RNA, and protein sequences and position correlation coefficients to generate feature vector encoding in SVM format. Parallel correlation pseudo amino acid composition (PC-PseAAC) is a method that combines local sequence-order information and global sequence-order information into a feature vector code of a protein sequence. Peptides have been indicated to be reliable drug candidates, and their use is still hindered mainly because of their high susceptibility towards protease degradation.

**HLP encoding.** In the past, a number of peptides have been reported to possess highly diverse properties ranging from cell penetrating, tumour homing, anticancer, anti-hypertensive, anti-viral to antimicrobials. Owing to their excellent specificity, low toxicity, rich chemical diversity and availability from natural sources, we used HLP (http://crdd.osdd.net/raghava/hlp/index.html) [26] as one of the feature coding methods. The life span of a protein is related not only to the folding structure and size of the protein but also to the protein action and cell state. HLP is a server developed for predicting the half-life of peptides in intestine-like environments. It generates all possible mutants (single mutation at each position per cycle) for a peptide and predicts/calculates the half-life and physicochemical properties (e.g., charge, polarity, hydrophobicity, volume, pK) of mutant peptides [27]. From the results page returned by the HLP server, because the input amino acid length is different, the number of cuts obtained is also different; so, we take the average of the 20 values on the results page as the code, including the half-life, stability, HPLC parameter, hydrophobicity, pKa, pKb, residue volume, molecular weight, isoelectric point, surface accessibility, flexibility, charge, polarity, relative mutability, free energy of solution, optical rotation, entropy of formation, heat capacity, and relative stability.

## Feature selection method

In this study, we used three different feature selected tools, including the LIBSVM feature selection tool [28], mRMR [29], and Weka attribute selection [30, 31]. Each feature selection method is different for their purpose in feature coding. Therefore, the results of the LIBSVM

feature selection tool and the results of mRMR were integrated and specifically ranked. The combined method was accumulated by the ranking of the feature numbers. For example, feature number 9 was ranked first in libsvm_fs, mRMR was ranked seventh, and its integration score was 1+7 = 8. Finally, the feature with a ranking score of 0 was removed, and the lower the integration score was, the higher the ranking was.

### Evaluation measures

To assess the predictive performance of the classifier, we used the formula below, where TP, FP, FN, and TN, which are true positives, false positives, false negatives, and true negatives, respectively. The sensitivity (Sn) of this type of protein oligomer reflects the percentage of correct predictions for that category. Specificity (Sp) on behalf of non-protein oligomers of this type indicates the percentage of correct predictions of non-class. Precision evaluates the correct rate of true positive data in forward-looking data. Accuracy (ACC) is used to assess the overall predictive power of the prediction accuracy. Matthews correlation coefficient (MCC) values range from −1 to 1, in which the value of 1 represents a completely correct prediction, the value of 0 represents random prediction, and the value of −1 represents exactly the opposite prediction, were used as well.

$$Sn = \frac{TP}{TP + FN}$$

$$Sp = \frac{TN}{TN + FP}$$

$$Precision = \frac{TP}{TP + FP}$$

$$ACC = \frac{TP + TN}{TP + FP + TN + FN}$$

$$MCC = \frac{(TP \times TN) - (FN \times FP)}{\sqrt{(TP + FN) \times (TN + FP) \times (TP + FP) \times (TN + FN)}}$$

### Flowchart and system construction

QUATgo is a two-layer architecture. First, a series of feature codes were encoded for each piece of data recorded by CD-Hit, and various subunits were later defined as one answer. Because the number of data with hetero decamers was too small, we chose the second small dataset as the basis of the training set. Fifty sequences from each subunit were used as the first layer classifier training set. The training data of the hetero-decamer were merged to hetero-more than dodecamer; so, we took 20 sequences from the hetero-decamer and 30 sequences from the hetero- more than dodecamer. The purpose of the second-tier architecture was to further analyse whether the final output was hetero-decamer or hetero- more than dodecamer, so we took 20 data from each of the two hetero-oligomers as a training set. The remaining data were serve as independent data. The two layers of the classifier used the different feature selection ranking to filter the best accuracy of the model. (Fig 1)

## Result and discussion

To find the best classifier for this issue, we compared a total of 11 different learning algorithms and evaluated the validity of the model with 10-fold cross validation. Including LIBSVM,

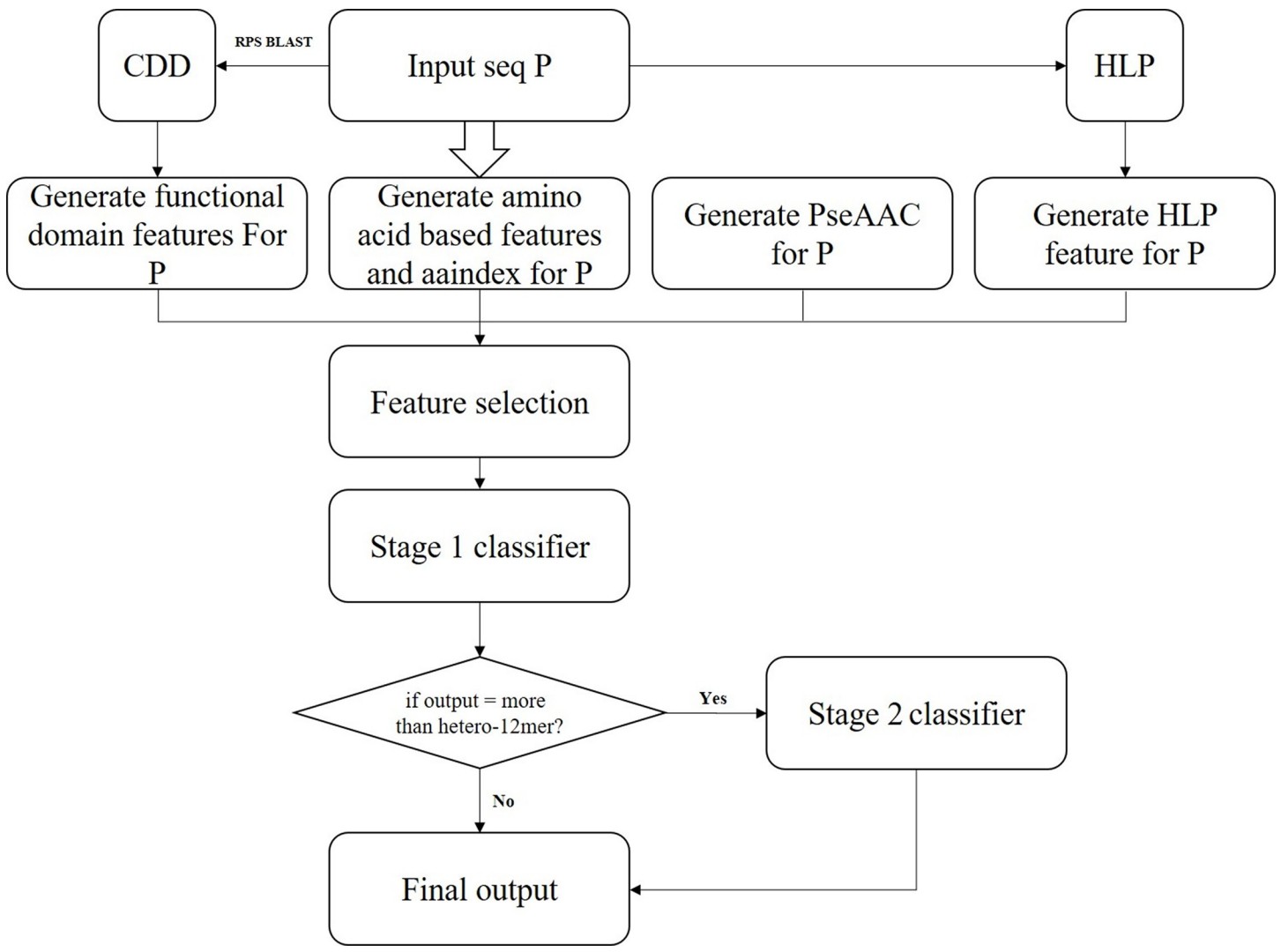

**Fig 1. System flow.**

developed by Professor Lin Chih-Jen of Taiwan University, Simple Logistic, Hoeffding, Random Forest, Random Tree, REP Tree, Decision Stump, DecisionTable, LogitBoost, J48, and OneR; these learning algorithms all belong to Weka.

We chose the top three classifiers for the next feature selection. Table 2 shows that the top three were Random Forest with 46.8% accuracy, Simple Logistic with 41.0% accuracy, and LogitBoost with 38.1% accuracy.

Then, the three classifiers were filtered through five different feature selection methods, as shown in Table 3. The best result was the Random Forest classifier, and the selection method was through Weka attribute selection. The final result was an accuracy of 49%, a total improvement of 2.2% over the total selection.

To further evaluate the effectiveness of the model, see Table 4, we further calculated the sensitivity, specificity, precision, accuracy (ACC), and MCC and evaluated the prediction model for each subunit. As shown in Table 4, for homo-oligomers with subunit numbers greater than 8 and in hetero-oligomers with subunit numbers greater than 6, the prediction accuracy of the above model was greater than the average accuracy. In the independent tests as Table 5, these

**Table 2. The cross-validation results of each step 1 classifier compared with different learning algorithms.**

| Algorithms | ACC |
|---|---|
| LIBSVM | 0.203 |
| Simple Logistic | 0.410 |
| Hoeffding | 0.343 |
| Random Forest | 0.468 |
| Random Tree | 0.351 |
| REPtree | 0.315 |
| Decision Stump | 0.101 |
| DecisionTable | 0.280 |
| LogitBoost | 0.381 |
| J48 | 0.350 |
| OneR | 0.243 |

oligomers have significant higher accuracy. This result indicates that the features left after feature selection have a greater correlation for the above subunits and that the data in the training set are easier to distinguish, such as homo-dimer, homo-trimer, homo-tetramer, homo-hexamer, hetero-dimer, hetero-trimer, hetero-tetramer units, probably because of the quantification of the coding that results in a low subunit number of protein oligomers that are not easy to predict. Since the ratio of the training data is to achieve for each subunit is 1:1, the total amount of subunits such as homo-dimers would be 2115, and only 50 data per training data may be selected for certain data volumes. This situation may be further improved by filtering the training set to achieve high accuracy.

In the second-level classifier, as shown in Table 6, there are seven classifiers with 95% accuracy. So we used these seven classifiers with different feature selection methods. The accuracy of two classifiers, Random Forest and LogitBoost, with feature selection methods in Weka attribute selection are both 100% (as shown in Table 7). For selecting the best model, the classifiers of Random Forest and LogiBoost were further tested in an independent dataset. Random Forest has better performance than LogiBoost (as shown in Table 8). Therefore, QUATgo uses Random Forest for model construction. On the other hand, there are three kinds of features were used in the first and second layer in the same time, the features are occurrence frequency of dipeptides, such as Proline (P) and Asparagine (N), dipeptide Serine (S) and Arginine (R), and Tyrosine (Y) and Threonine (T). However, there are not relevant data that can confirm this phenomenon in academic papers yet. Perhaps it can be used as a starting point for structural exploration and further verification using structural biology experiments. Most worthy of mention is the five kinds of features were used in QUATgo model construction, the feature selection algorithm reserved some coding for each feature. In other words, the features we adopted are useful for protein quaternary structural attributes prediction.

**Table 3. The cross-validation results of each classifier in stage 1 with different feature selection methods.**

| Selection method/classifier | Simple Logistic | LogitBoost | Random Forest |
|---|---|---|---|
| libsvm_fs | 0.388 | 0.379 | 0.486 |
| mid_maxrel | 0.373 | 0.383 | 0.458 |
| mid_mrmr | 0.235 | 0.340 | 0.391 |
| miq_maxrel | 0.373 | 0.383 | 0.458 |
| miq_mrmr | 0.391 | 0.388 | 0.485 |
| all_fs_rk | 0.479 | 0.376 | 0.386 |
| Weka attribute selection | 0.368 | 0.371 | 0.490 |

**Table 4. Predicting results of the best classifier of the stage 1 classifier with cross verification.**

|  | Sn | Sp | Precision | ACC | MCC |
|---|---|---|---|---|---|
| Monomer | 0.480 | 0.927 | 0.304 | 0.899 | 0.330 |
| Homo-2 mer | 0.320 | 0.945 | 0.281 | 0.906 | 0.250 |
| Homo-3 mer | 0.220 | 0.945 | 0.212 | 0.900 | 0.162 |
| Homo-4 mer | 0.060 | 0.960 | 0.091 | 0.904 | 0.024 |
| Homo-6 mer | 0.360 | 0.955 | 0.346 | 0.918 | 0.309 |
| Homo-8 mer | 0.480 | 0.971 | 0.522 | 0.940 | 0.469 |
| Homo-10 mer | 0.760 | 0.980 | 0.717 | 0.966 | 0.720 |
| Homo-12 mer | 0.660 | 0.975 | 0.635 | 0.955 | 0.623 |
| Homo-mt12 mer* | 0.760 | 0.992 | 0.864 | 0.978 | 0.798 |
| Hetero-2 mer | 0.540 | 0.937 | 0.365 | 0.913 | 0.399 |
| Hetero-3 mer | 0.360 | 0.955 | 0.346 | 0.918 | 0.309 |
| Hetero-4 mer | 0.140 | 0.959 | 0.184 | 0.908 | 0.112 |
| Hetero-6 mer | 0.420 | 0.979 | 0.568 | 0.944 | 0.459 |
| Hetero-8 mer | 0.620 | 0.995 | 0.886 | 0.971 | 0.727 |
| Hetero-12 mer | 0.780 | 0.993 | 0.886 | 0.980 | 0.821 |
| Hetero-mt12 mer* | 0.880 | 0.989 | 0.846 | 0.983 | 0.854 |
| Avg | 0.490 | 0.966 | 0.503 | 0.936 | 0.460 |

* = More than 12 mer

## Comparing with different prediction tools

We compared QUATgo with other quaternary structure prediction tools by independent test set, and calculated precision, sensitivity, specificity, accuracy (ACC), and MCC. Because some categories in the independent test set have a small amount of data, in order not to let the data

**Table 5. Predicting results of best classifier of stage 1 classifier with independent testing.**

|  | Sn | Sp | Precision | ACC | MCC |
|---|---|---|---|---|---|
| Monomer | 0.472 | 0.818 | 0.381 | 0.752 | 0.269 |
| Homo-2 mer | 0.218 | 0.882 | 0.304 | 0.755 | 0.114 |
| Homo-3 mer | 0.271 | 0.922 | 0.278 | 0.856 | 0.195 |
| Homo-4 mer | 0.154 | 0.955 | 0.450 | 0.802 | 0.174 |
| Homo-6 mer | 0.245 | 0.952 | 0.239 | 0.911 | 0.195 |
| Homo-8 mer | 0.483 | 0.951 | 0.320 | 0.929 | 0.358 |
| Homo-10 mer | 0.667 | 0.990 | 0.071 | 0.990 | 0.216 |
| Homo-12 mer | 0.500 | 0.963 | 0.202 | 0.954 | 0.298 |
| Homo-mt12 mer* | 0.750 | 0.987 | 0.353 | 0.985 | 0.508 |
| Hetero-2 mer | 0.285 | 0.962 | 0.417 | 0.903 | 0.294 |
| Hetero-3 mer | 0.370 | 0.936 | 0.093 | 0.926 | 0.158 |
| Hetero-4 mer | 0.104 | 0.950 | 0.081 | 0.916 | 0.048 |
| Hetero-6 mer | 0.512 | 0.981 | 0.304 | 0.974 | 0.382 |
| Hetero-8 mer | 1.000 | 0.988 | 0.030 | 0.988 | 0.173 |
| Hetero-12 mer | 0.893 | 0.993 | 0.595 | 0.992 | 0.726 |
| Hetero mt12 mer* | 0.776 | 0.995 | 0.731 | 0.990 | 0.748 |
| Avg | 0.481 | 0.951 | 0.303 | 0.914 | 0.303 |

* = More than 12 mer

**Table 6. The cross-validation results of each stage 2 classifier with each comparing learning algorithms.**

| Algorithms | ACC |
|---|---|
| LIBSVM | 0.700 |
| Simple Logistic | 0.925 |
| Hoeffding | 0.950 |
| Random Forest | 0.950 |
| Random Tree | 0.950 |
| REPtree | 0.950 |
| Decision Stump | 0.950 |
| DecisionTable | 0.925 |
| LogitBoost | 0.950 |
| J48 | 0.950 |
| OneR | 0.900 |

**Table 7. The cross-validation results of each stage 2 classifier with different feature selection methods.**

| 10-fold cross-validation | Hoeffding | Random Forest | Random Tree | REPtree | Decision stump | LogitBoost | J48 |
|---|---|---|---|---|---|---|---|
| libsvm_fs | 0.95 | 0.925 | 0.9 | 0.9 | 0.85 | 0.9 | 0.875 |
| mid_maxrel | 0.925 | 0.95 | 0.95 | 0.975 | 0.95 | 0.95 | 0.95 |
| mid_mrmr | 0.95 | 0.925 | 0.925 | 0.95 | 0.95 | 0.925 | 0.95 |
| miq_maxrel | 0.925 | 0.95 | 0.95 | 0.975 | 0.95 | 0.95 | 0.95 |
| miq_mrmr | 0.95 | 0.95 | 0.925 | 0.975 | 0.975 | 0.975 | 0.95 |
| all_fs_rk | 0.95 | 0.95 | 0.95 | 0.975 | 0.975 | 0.975 | 0.95 |
| Weka attribute selection | 0.95 | 1 | 0.95 | 0.975 | 0.975 | 1 | 0.975 |

**Table 8. Predicting results of the best classifier of stage 2 classifiers with cross-validation and independent testing.**

| Independent testing | Random Forest | LogitBoost |
|---|---|---|
| Sn | 0.959 | 0.796 |
| Sp | - | - |
| Precision | 1.000 | 1.000 |
| ACC | 0.959 | 0.796 |
| MCC | - | - |

of a single category excessively affect the overall accuracy, a maximum of 100 pieces of data are selected in each category for testing. In Table 9, the overall average MCC of QUATgo is higher than other tools, and the MCC is greater than zero in each category. Because there is no data predicted as homo-decamer, the precision and MCC of QuarIdent in the homo-decamer test set are not applicable (N/A). Besides, QuarBingo also shows the non-applicable results in three categories.

## Case study

To understand the ability of QUATgo to predict the quaternary structure of different functional proteins, we conducted a case study using influenza virus proteins. The influenza virus is an RNA virus belonging to the orthomyxoviridae family [32]. According to its

**Table 9. Comparing other prediction tools.**

| mer | Precision | | | Sn | | | Sp | | | ACC | | | MCC | | |
|---|---|---|---|---|---|---|---|---|---|---|---|---|---|---|---|
| | QuarIdent | QUATgo | QuarBingo | QuarIdent | QUATgo | QuarBingo | QuarIdent | QUATgo | QuarBingo | QuarIdent | QUATgo | QuarBingo | QuarIdent | QUATgo | QuarBingo |
| Monomer | 0.438 | 0.331 | 0.143 | 0.320 | 0.430 | 0.880 | 0.953 | 0.900 | 0.390 | 0.887 | 0.851 | 0.441 | 0.314 | 0.294 | 0.171 |
| Homo-2 mer | 0.224 | 0.223 | 0.221 | 0.520 | 0.270 | 0.190 | 0.792 | 0.892 | 0.923 | 0.764 | 0.827 | 0.847 | 0.223 | 0.149 | 0.121 |
| Homo-3 mer | 0.313 | 0.260 | 0.641 | 0.310 | 0.260 | 0.410 | 0.922 | 0.915 | 0.973 | 0.858 | 0.847 | 0.915 | 0.233 | 0.175 | 0.470 |
| Homo-4 mer | 0.452 | 0.278 | 0.588 | 0.380 | 0.200 | 0.100 | 0.947 | 0.940 | 0.992 | 0.888 | 0.863 | 0.900 | 0.353 | 0.162 | 0.213 |
| Homo-6 mer | 0.689 | 0.313 | 0.500 | 0.310 | 0.250 | 0.010 | 0.984 | 0.937 | 0.999 | 0.914 | 0.866 | 0.897 | 0.425 | 0.206 | 0.059 |
| Homo-8 mer | 0.887 | 0.686 | N/A | 0.550 | 0.350 | 0.000 | 0.992 | 0.982 | 1.000 | 0.946 | 0.916 | 0.897 | 0.674 | 0.452 | N/A |
| Homo-10 mer | N/A | 0.105 | N/A | 0.000 | 0.667 | 0.000 | 1.000 | 0.982 | 1.000 | 0.997 | 0.981 | 0.997 | N/A | 0.260 | N/A |
| Homo-12 mer | 1.000 | 0.500 | 0.263 | 0.604 | 0.500 | 0.750 | 1.000 | 0.974 | 0.890 | 0.980 | 0.950 | 0.883 | 0.769 | 0.474 | 0.399 |
| Hetero-2 mer | 0.386 | 0.387 | 0.538 | 0.220 | 0.290 | 0.140 | 0.960 | 0.947 | 0.986 | 0.883 | 0.879 | 0.899 | 0.232 | 0.270 | 0.237 |
| Hetero-3 mer | 0.400 | 0.174 | 0.333 | 0.043 | 0.348 | 0.022 | 0.997 | 0.917 | 0.998 | 0.951 | 0.890 | 0.951 | 0.119 | 0.192 | 0.075 |
| Hetero-4 mer | 0.429 | 0.193 | 0.333 | 0.060 | 0.110 | 0.010 | 0.991 | 0.947 | 0.998 | 0.895 | 0.860 | 0.896 | 0.129 | 0.074 | 0.042 |
| Hetero-6 mer | 0.000 | 0.391 | 0.250 | 0.000 | 0.439 | 0.024 | 0.997 | 0.970 | 0.997 | 0.954 | 0.947 | 0.956 | -0.012 | 0.387 | 0.066 |
| Hetero-8 mer | 0.000 | 0.063 | N/A | 0.000 | 1.000 | 0.000 | 0.999 | 0.984 | 1.000 | 0.998 | 0.984 | 0.999 | -0.001 | 0.248 | N/A |
| Hetero-12 mer | 0.000 | 0.813 | 0.000 | 0.000 | 0.929 | 0.000 | 0.998 | 0.994 | 0.995 | 0.969 | 0.992 | 0.966 | -0.008 | 0.864 | -0.012 |
| Avg | 0.401 | 0.337 | 0.346 | 0.237 | 0.432 | 0.181 | 0.966 | 0.949 | 0.939 | 0.920 | 0.904 | 0.889 | 0.265 | 0.301 | 0.167 |

nucleoproteins, the influenza virus is classified as genera A, B, C and D [33]. Influenza viruses infect people, pigs, horses, and poultry. Influenza A infects a variety of species is zoonotic and influenza B infects only humans, however, both of them often cause seasonal influenza. Influenza A can be further classified according to the surface hemagglutinin and neuraminidase. Hemagglutinin has 18 subtypes [33] that are responsible for identifying and binding to host cell membranes. Neuraminidase has 11 subtypes that destroy host receptor proteins. Influenza viruses migrate from host to host through hemagglutinin mutations, and it is not yet possible to accurately predict the virus subtypes that will become prevalent. We collected seven subtypes of hemagglutinin protein sequences for quaternary structure prediction, with PDB identifier 1RUZ, 2FK0, 2IBX, 2VIU, 3BT6, 3EYJ, 3ZNM, 4BSG, 4D00, 4UO0, 4UO4, 5HMG and 6CF7, respectively. The sequence identity between the 13 protein sequences and our train data set is less than 40%. The QUATgo prediction results showed that the classification accuracy of the 13 quaternary structures was 61.54% (8/13), while the other two tools, QuatIdent and Qua-Bingo, could not correctly predict the quaternary structures.

## Conclusions and future work

Proteins contain both single-chain proteins and multi-chain proteins. According to the number of constituent chains, proteins can be classified into many different quaternary structural attributes, such as monomer, dimer, and trimer. Each of these oligomers can be further classified as a homo-oligomer formed by identical chains (subunits) or a hetero-oligomer by different chains. These properties are closely related to the function of the protein. For example, some ligands bind only to dimers, but not to monomers, and some exotic allosteric transitions occur only in homotetramers, such as the CArG-box gene region in the MADS-box gene of plants.

Proteins composed of different numbers of protein quaternary structural subunits reflect the evolution of structure and function in protein composition. Therefore, it is of great significance to use the hybrid feature encoding method to complete the prediction of protein quaternary structure.

This study proposes the use of mixed feature coding to predict the quaternary structure of proteins. Although the overall prediction result in the independent set was better than other tools, there is potential for predicting lower subunit sets. Because numerically coded protein sequences may have different protein sequences, they have similar feature codes, resulting in misjudgement of the prediction system. This result is reflected in homo-dimer, homo-trimer, homo-tetramer, homo-hexamer, hetero-dimer, hetero-trimer, and hetero-tetramer units; however, there is a minimum of 91.6% accuracy to a maximum of 99.2% accuracy in homo-octamer, homo-decamer, homo-dodecamer, hetero-hexamer, hetero-octamer, hetero-dodeca-mer. The current prediction tools only predict the dodecahedron but the proteins above the dodecamer. Based on the results of this study, it is feasible to use this hybrid feature encoding method to predict protein oligomers with high subunit numbers.

Many proteins exist as monomers whether they interact with another protein to form polymers, and whether the protein dimer will be further assembled into biologically relevant tetramers or octamers. Currently, most of these problems have not been addressed by scientific research and verified by enough information. In the future, more and more data will be added to the database, providing more useful information to establish a prediction system and assist relevant research development.

## Acknowledgments

The authors would like to thank Professor Jyung-Hurng Liu who provided the suggestions for case study.

## Author Contributions

**Conceptualization:** Chi-Hua Tung, Yen-Wei Chu.

**Formal analysis:** Lan-Ying Huang, Yu-Nan Liu.

**Investigation:** Yu-Nan Liu.

**Methodology:** Chi-Wei Chen.

**Project administration:** Chi-Hua Tung, Yen-Wei Chu.

**Software:** Ching-Hsuan Chien.

**Supervision:** Chi-Hua Tung, Yen-Wei Chu.

**Validation:** Ching-Hsuan Chien, Chi-Wei Chen, Lan-Ying Huang.

**Visualization:** Ching-Hsuan Chien.

**Writing – original draft:** Chi-Hua Tung, Ching-Hsuan Chien, Yu-Nan Liu, Yen-Wei Chu.

**Writing – review & editing:** Chi-Hua Tung, Ching-Hsuan Chien, Yen-Wei Chu.

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
