## [Decision Letter · Decision Letter 0]

4 Sep 2019

PONE-D-19-21078

QUATgo: Protein quaternary structural attributes prediction by two-stage machine learning approaches with heterogeneous feature encoding

PLOS ONE

Dear Prof. Chu,

Thank you for submitting your manuscript to PLOS ONE. After careful consideration, we feel that it has merit but does not fully meet PLOS ONE’s publication criteria as it currently stands. Therefore, we invite you to submit a revised version of the manuscript that addresses the points raised during the review process.

We would appreciate receiving your revised manuscript by Oct 19 2019 11:59PM. To enhance the reproducibility of your results, we recommend that if applicable you deposit your laboratory protocols in protocols.io, where a protocol can be assigned its own identifier (DOI) such that it can be cited independently in the future. For instructions see: http://journals.plos.org/plosone/s/submission-guidelines#loc-laboratory-protocols

We look forward to receiving your revised manuscript.

Kind regards,

Mingyang Lu

Academic Editor

PLOS ONE

Journal Requirements:

Additional Editor Comments (if provided):

As mentioned in the referee reports, both reviewers pointed that the manuscript is not presented well in English. It is also not acceptable that sentences are directly copied from Wikipedia. Moreover, there are concerns about the benchmark test. I suggest the authors carefully address these questions before the manuscript can be considered for publication in PLOS ONE.

Reviewers' comments:

Reviewer's Responses to Questions

**Comments to the Author**

1. Is the manuscript technically sound, and do the data support the conclusions?

Reviewer #1: Yes

Reviewer #2: No

2. Has the statistical analysis been performed appropriately and rigorously? 

Reviewer #1: Yes

Reviewer #2: No

3. Have the authors made all data underlying the findings in their manuscript fully available?

Reviewer #1: Yes

Reviewer #2: No

4. Is the manuscript presented in an intelligible fashion and written in standard English?

Reviewer #1: No

Reviewer #2: No

5. Review Comments to the Author

Reviewer #1: The authors employ hybrid feature encoding method to propose a new tool (QUATgo) for the prediction of the quaternary structure of proteins. This tool give thought to the integration of heterogeneous coding and the accuracy of subunit categories with low data number.

This is an important endeavor to overcoming the inadequacies of current predicting methods. Accordingly, the proposed method has potential to predict the protein oligomers with high subunit numbers.

This work is very interesting. I therefore recommend publication in the PLOS ONE. I have some minor comments that the authors can address:

Writing throughout needs work.

The definition of "quaternary structure" repeated many times in introduction.

Some sentences in the introduction are directly copied from Wikipedia.

Introduction: " In recent years, Shen et al. have also proposed coding methods based on 71 Functional Domain Composition" lacks a citation.

Reviewer #2: This MS does not show it is superior to existing method. Looking at Table 9 on page 15, for many

of the categories, its precision is smaller than that of "QuarIdent". Actually, this kind of comparison

is meaningless because we do not know many possible overlaps of the testing set with the training set and do not know how similar is the testing set to the training set. I do not even know how the "independent test set" is constructed other than a sentence saying "We compared QUATgo with other quaternary structure prediction tools. We use the remaining files in the training set as independent sets". The fact that this MS's method is doing well on the larger "MERS" could be due to that there are high sequence identity sequences in the training set to the testing set because for these "MERS", the high sequence identity ones are not removed by CD-hit.

The MS is written in poor English. There are many sentences like these:

"The quaternary structure of the homo-oligomer can be encoded by the Equation an, where n is the copy number of the polypeptide chain."

"If the protein sequence has similar functional domains, it may represent evolutionarily related

[8, 9]. If there is a similar functional domain in the protein sequence, it may be evolutionarily related."

"Comparing similarities with RPS-BLAST through the Conserved Domains database (CDD) [10]."

"Proteins of different functions, sources and species, in addition to structural information verification through time-consuming experimental methods,if the homology between the sequences is low, it is more difficult to find the protein aggregation rules by using the alignment method."

......

6. PLOS authors have the option to publish the peer review history of their article (what does this mean?). If published, this will include your full peer review and any attached files.

Reviewer #1: No

Reviewer #2: No

---

## [Author Response · Author response to Decision Letter 0]

15 Oct 2019

I. Answer to comments from Reviewer #1:

1. Writing throughout needs work.

Response: 

Thank you for your advice, and we appreciate the chance to make this manuscript clear. We have our manuscripts checked by English language native speakers for English editing. The correctness of English writing has been improved. In addition, the English revision certificate has been uploaded as an attachment.

2. The definition of "quaternary structure" repeated many times in introduction.

Some sentences in the introduction are directly copied from Wikipedia.

Response: 

Thanks for your comment. According to your suggestion, we have rewritten the first paragraph of the introduction. We restate the definition of quaternary structure of protein and the importance of the quaternary structure prediction research while avoiding plagiarism.

3. Introduction: "In recent years, Shen et al. have also proposed coding methods based on 71 Functional Domain Composition" lacks a citation.

Response: Thank you for reminding. We have added the citation (reference number: [1]) correctly.

II. Answer to comments from Reviewer #2:

1. This MS does not show it is superior to existing method. Looking at Table 9 on page 15, for many of the categories, its precision is smaller than that of "QuarIdent".

Response: 

Thanks for your comment. Although some results were worse than QuarIdent in precision, the overall average of QUATgo was better than QuarIdent. At the same time, we mainly use MCC as the evaluation index, because it can take into account both true and false positives and negatives and is generally regarded as a balanced measure which can be used even if the classes are of very different sizes. For each independent test, QUATgo was superior to the other two existing methods in MCC results. And, the overall average ACC and MCC results of QUATgo are also higher than the other two methods.

2. Actually, this kind of comparison is meaningless because we do not know many possible overlaps of the testing set with the training set and do not know how similar is the testing set to the training set.

Response: 

Thanks for your comment. These two data sets do not have the same or similar protein family. In the homo-oligomers and hetero-oligomers data set, we compare the testing set and the training set where 95% of the sequence data in testing set has sequence identity lower than 40%.

3. I do not even know how the "independent test set" is constructed other than a sentence saying "We compared QUATgo with other quaternary structure prediction tools. We use the remaining files in the training set as independent sets".

Response:

Thank you for your advice, and we appreciate the chance to make this manuscript clear. In revised manuscript, we rewrite two parts of statements as follows. 

. Adding the sentence “The remaining data were serve as independent data.” In the section “Flowchart and system construction” to show how the independent test set was made.

. Rewriting the sentences of “We compared QUATgo with other quaternary structure prediction tools by independent test set, and calculated precision, sensitivity, specificity, accuracy (ACC), and MCC.” In the section “Comparing with different prediction tools”

4. The fact that this MS's method is doing well on the larger "MERS" could be due to that there are high sequence identity sequences in the training set to the testing set because for these "MERS", the high sequence identity ones are not removed by CD-hit.

Response: 

Thanks for your comment. We further inspected the sequence identity of these MERS S ectodomains to our training set. The results showed that the sequence identity of these hetero-12 mer are about 60%. Please see the table below for detailed results.

PDBID Identity

5W9H 62.16%

5W9I 59.81%

5W9J 59.81%

5W9K 60%

5W9L 59.81%

5W9M 62.16%

5W9N 62.16%

5W9O 62.16%

5W9P 61.26%

5W9Q <10%

5. The MS is written in poor English. There are many sentences like these:

"The quaternary structure of the homo-oligomer can be encoded by the Equation an, where n is the copy number of the polypeptide chain."

"If the protein sequence has similar functional domains, it may represent evolutionarily related

[8, 9]. If there is a similar functional domain in the protein sequence, it may be evolutionarily related."

"Comparing similarities with RPS-BLAST through the Conserved Domains database (CDD) [10]."

"Proteins of different functions, sources and species, in addition to structural information verification through time-consuming experimental methods,if the homology between the sequences is low, it is more difficult to find the protein aggregation rules by using the alignment method."

......

Response: 

Thank you for your advice, and we appreciate the chance to make this manuscript clear. We have our manuscripts checked by English language native speakers for English editing. The correctness of English writing has been improved. In addition, the English revision certificate has been uploaded as an attachment.

"The quaternary structure of the homo-oligomer can be encoded by the Equation an, where n is the copy number of the polypeptide chain." This sentence has been deleted because of its inappropriateness.

"If the protein sequence has similar functional domains, it may represent evolutionarily related [8, 9]. If there is a similar functional domain in the protein sequence, it may be evolutionarily related." These sentences have been re-written to “If the protein sequence has similar functional domains, it may represent an evolutionarily relationship [8, 9].”, and deleted another inappropriate sentence.

"Comparing similarities with RPS-BLAST through the Conserved Domains database (CDD) [10]." This sentences has been re-written to “Comparing similarities can be performed with RPS-BLAST through the Conserved Domain Database (CDD) [10].”

"Proteins of different functions, sources and species, in addition to structural information verification through time-consuming experimental methods,if the homology between the sequences is low, it is more difficult to find the protein aggregation rules by using the alignment method." This sentences has been re-written to “For proteins of different functions, sources and species, in addition to verifying structural information, it is more difficult to find the protein aggregation rules using the alignment method if the homology between the sequences is low.”

---

## [Decision Letter · Decision Letter 1]

8 Nov 2019

PONE-D-19-21078R1

QUATgo: Protein quaternary structural attributes predicted by two-stage machine learning approaches with heterogeneous feature encoding

PLOS ONE

Dear Prof. Chu,

Thank you for submitting your manuscript to PLOS ONE. After careful consideration, we feel that it has merit but does not fully meet PLOS ONE’s publication criteria as it currently stands. Therefore, we invite you to submit a revised version of the manuscript that addresses the points raised during the review process.

We would appreciate receiving your revised manuscript by Dec 23 2019 11:59PM. To enhance the reproducibility of your results, we recommend that if applicable you deposit your laboratory protocols in protocols.io, where a protocol can be assigned its own identifier (DOI) such that it can be cited independently in the future. For instructions see: http://journals.plos.org/plosone/s/submission-guidelines#loc-laboratory-protocols

We look forward to receiving your revised manuscript.

Kind regards,

Mingyang Lu

Academic Editor

PLOS ONE

Reviewers' comments:

Reviewer's Responses to Questions

**Comments to the Author**

1. If the authors have adequately addressed your comments raised in a previous round of review and you feel that this manuscript is now acceptable for publication, you may indicate that here to bypass the “Comments to the Author” section, enter your conflict of interest statement in the “Confidential to Editor” section, and submit your "Accept" recommendation.

Reviewer #1: All comments have been addressed

Reviewer #2: (No Response)

2. Is the manuscript technically sound, and do the data support the conclusions?

Reviewer #1: Yes

Reviewer #2: Partly

3. Has the statistical analysis been performed appropriately and rigorously? 

Reviewer #1: Yes

Reviewer #2: No

4. Have the authors made all data underlying the findings in their manuscript fully available?

Reviewer #1: Yes

Reviewer #2: Yes

5. Is the manuscript presented in an intelligible fashion and written in standard English?

Reviewer #1: Yes

Reviewer #2: Yes

6. Review Comments to the Author

Reviewer #1: I carefully read the revised manuscript and the response to all reviewer concerns. I think the authors successfully responded to my concerns and recommend now acceptance of the manuscript.

Reviewer #2: A major issue with the MS is the error in MCC calculation of their QUATgo in Table 9. Based on a quick check on Homo-12-mer, QuarIdent has a much larger precision (0.939 vs. 0.271), close specificity (0.998 vs. 0.999), and slightly worse sensitivity (0.018 vs. 0.030), yet its MCC value is much worse than that of QUATgo (0.067 vs. 0.990 !). This does not make sense. In fact, based on the values of precision, sn, sp, I can infer its MCC value is 0.0866846 that is slightly better than 0.067 of QuarIdent. Many other MCC values are also wrong, i.e. Monomer: MCC=-0.103 instead of 0.378;Homo-2:MCC=-0.123; Homo-3: MCC=-0.182 ... On average, their MCC value is worse than that of QuarIdent ! Unless the authors correctly calculate their MCC and show they are statistically significantly better than other methods (i.e. p-value < 0.05 by student-t test), I do not see the value of publishing this work.

7. PLOS authors have the option to publish the peer review history of their article (what does this mean?). If published, this will include your full peer review and any attached files.

Reviewer #1: No

Reviewer #2: No

---

## [Author Response · Author response to Decision Letter 1]

23 Mar 2020

I. Answer to comments from Reviewer #1:

I carefully read the revised manuscript and the response to all reviewer concerns. I think the authors successfully responded to my concerns and recommend now acceptance of the manuscript.

Response: 

Thank you for your advice.

II. Answer to comments from Reviewer #2:

A major issue with the MS is the error in MCC calculation of their QUATgo in Table 9. Based on a quick check on Homo-12-mer, QuarIdent has a much larger precision (0.939 vs. 0.271), close specificity (0.998 vs. 0.999), and slightly worse sensitivity (0.018 vs. 0.030), yet its MCC value is much worse than that of QUATgo (0.067 vs. 0.990 !). This does not make sense. In fact, based on the values of precision, sn, sp, I can infer its MCC value is 0.0866846 that is slightly better than 0.067 of QuarIdent. Many other MCC values are also wrong, i.e. Monomer: MCC=-0.103 instead of 0.378;Homo-2:MCC=-0.123; Homo-3: MCC=-0.182 ... On average, their MCC value is worse than that of QuarIdent ! Unless the authors correctly calculate their MCC and show they are statistically significantly better than other methods (i.e. p-value < 0.05 by student-t test), I do not see the value of publishing this work.

Response: 

Thank you very much for your suggestions. We found that there was a bias towards negative in the QUATgo system. It shows that the specificity (sp) was exceedingly high and the sensitivity (sn) was too low. Therefore, we spent an extra three months to re-adjust the test set and revise it to the ratio of positive and negative set equal to 1:1. Finally, accuracy and reliability are improved, and the results will not be excessively biased towards sp. Due to the update of the test data set, there are many corrections in the table and the article, which we have highlighted in yellow. Besides, we modify the cases discussed in the case study. We used the Influenza hemagglutinin protein as our case study. The results showed that 8 of 13 hemagglutinin proteins correctly predicted their quaternary structure (hetero 6-mer), while the QuatIdent and QuaBingo were all wrong. Moreover, the sequence identity between these 13 hemagglutinins and our train data set is less than 40%. Thanks for your reminding, let us be able to improve and make the research results more precise.

---

## [Decision Letter · Decision Letter 2]

8 Apr 2020

QUATgo: Protein quaternary structural attributes predicted by two-stage machine learning approaches with heterogeneous feature encoding

PONE-D-19-21078R2

Dear Dr. Chu,

We are pleased to inform you that your manuscript has been judged scientifically suitable for publication and will be formally accepted for publication once it complies with all outstanding technical requirements.

With kind regards,

Mingyang Lu

Academic Editor

PLOS ONE

Additional Editor Comments (optional):

Reviewers' comments:

Reviewer's Responses to Questions

**Comments to the Author**

1. If the authors have adequately addressed your comments raised in a previous round of review and you feel that this manuscript is now acceptable for publication, you may indicate that here to bypass the “Comments to the Author” section, enter your conflict of interest statement in the “Confidential to Editor” section, and submit your "Accept" recommendation.

Reviewer #2: All comments have been addressed

2. Is the manuscript technically sound, and do the data support the conclusions?

Reviewer #2: Yes

3. Has the statistical analysis been performed appropriately and rigorously? 

Reviewer #2: Yes

4. Have the authors made all data underlying the findings in their manuscript fully available?

Reviewer #2: Yes

5. Is the manuscript presented in an intelligible fashion and written in standard English?

Reviewer #2: Yes

6. Review Comments to the Author

Reviewer #2: (No Response)

7. PLOS authors have the option to publish the peer review history of their article (what does this mean?). If published, this will include your full peer review and any attached files.

Reviewer #2: No

---

## [Editor Report · Acceptance letter]

13 Apr 2020

PONE-D-19-21078R2 

QUATgo: Protein quaternary structural attributes predicted by two-stage machine learning approaches with heterogeneous feature encoding 

Dear Dr. Chu:

I am pleased to inform you that your manuscript has been deemed suitable for publication in PLOS ONE. Congratulations! Your manuscript is now with our production department. 

With kind regards,

on behalf of

Dr. Mingyang Lu 

Academic Editor

PLOS ONE